# Prediction of Ki67 scores from H&E stained breast cancer sections using convolutional neural networks

**Philippe Weitz**[1]                                                                PHILIPPE.WEITZ@KI.SE

**Balazs Acs**[2,3]                                                                     BALAZS.ACS@KI.SE

**Johan Hartman**[2,3]                                                              JOHAN.HARTMAN@KI.SE

**Mattias Rantalainen**[1]                                               MATTIAS.RANTALAINEN@KI.SE

[1] *Department of Medical Epidemiology and Biostatistics, Karolinska Institutet, Stockholm, Sweden*

[2] *Department of Oncology-Pathology, Karolinska Institutet, Stockholm, Sweden*

[3] *Department of Clinical Pathology and Cytology, Karolinska University Laboratory*

**Editors:** Under Review for MIDL 2021

## Abstract

Ki67 is an established marker of proliferation in breast cancer, but currently has limited clinical value due to limitations of the analytical validity of immunohistochemistry (IHC) - based Ki67 scoring. While the inter-assessor variability of scoring can be improved through image analysis software, Ki67 IHC also suffers from a lack of standardized staining protocols and is not part of routine pathology workflow in most countries. This could potentially be alleviated through directly predicting Ki67 scores from routine hematoxylin and eosin (H&E) stained whole-slide-images (WSIs). We compared four different deep learning based approaches to predict Ki67 scores from routine H&E stained WSIs in a dataset that consists of matched H&E and Ki67 WSIs from 126 breast cancer patients, resulting in a Spearman correlation between WSI cancer ROI averages of 0.546 for the best performing model in a 5-fold cross-validation (CV). These findings suggest that it is possible to predict the Ki67 score from H&E stained WSIs, but validation in a larger cohort is required to meaningfully distinguish the performance of the methods that were investigated.

**Keywords:** Ki67, breast cancer, registration, stain transformation, Cycle-GAN

## 1. Introduction

Ki67 immunohistochemistry staining (IHC) is a marker of proliferation in breast cancer, in the clinic in can provide information for treatment decisions relating to adjuvant chemotherapy in early stage ER-positive cancers. However, the clinical utility of Ki67 scoring is currently limited by its analytical validity, which requires robust standards both for scoring and sample preparation (Nielsen et al., 2020). Ki67 scoring consists of counting stain status of a large number of cells and computing the percentage of Ki67 positive cells in a cancer ROI. While the International Ki67 in Breast Cancer Working Group (IKWG) has thoroughly investigated how to standardize scoring, e.g. with the open source software QuPath (Bankhead et al., 2017; Acs et al., 2020), there are still no clear guidelines for sample preparation, including staining platforms, methods for antigen retrieval and Ki67 antibodies and counterstains (Nielsen et al., 2020). Variations in IHC sample preparation could be eliminated entirely by directly predicting Ki67 scores from standardized and ubiquitously available routine H&E stained sections. It has recently been demonstrated that

CNNs can predict a wide range of clinical and molecular characteristics in breast cancer (Couture et al., 2018). Previous studies also indicated that Cycle-GANs may be able to perform unpaired domain translation between different histological stainings (Zhu et al., 2017; Gupta et al., 2018). In this study, we compared four different modelling approaches to evaluate whether predicting Ki67 scores with CNNs from WSIs of H&E stained sections can yield robust estimates of the Ki67 cancer ROI average score.

## 2. Methods

This study includes WSIs of paired Ki67 and H&E stained sections of 126 female breast cancer patients from the Clinseq study. Ki67 scores for each IHC WSI were generated by a trained pathologist using QuPath and the percentage of Ki67-positive cells across the cancer ROI was used as the WSI label. All WSIs were tiled at 20X (0.4536µm/pixel) with a tile size and stride of 500 pixels. Patients were randomly assigned to 5 cross-validation (CV) folds for model evaluation. In the first modelling approach, we assigned the Ki67 ROI average as a weak label to each H&E tile of a WSI. We compared this with a registration-based approach that uses local QuPath cell detections. To this end, we used an elastic registration algorithm (Wodzinski and Skalski, 2021). To increase the robustness of the registration against artefacts in the WSIs, we modified it such that only SIFT and SURF features from the H&E and Ki67 cancer ROIs are used for the initial rigid registration. After transforming the coordinates of all detected cells with the resulting deformation field, we computed the percentage of Ki67-positive cells for each H&E tile to obtain local labels. We furthermore compared this to two different Cycle-GAN-based modelling approaches (Zhu et al., 2017). In the first approach, we trained regression CNNs to predict the percentage of Ki67-positive cells for each tile using IHC tiles. To predict Ki67 scores from H&E WSIs, we transformed H&E tiles to the IHC domain. In the second approach, we transformed all IHC tiles to the H&E domain and trained regression CNNs with the generated images, again using the cell detections from QuPath to compute a percentage of Ki67-positive cells for each tile.

## 3. Results & Discussion

The four suggested modelling approaches were compared in a 5-fold CV, with Spearman correlation between predicted and IHC determined K67 score as ROI averages as the primary performance metric. This results in the correlations shown in Figure 1, with a correlation of 0.527 (bootstrapped 95% CI 0.387, 0.648) for training with weak labels, 0.546 (0.409, 0.66) for training with registered labels, 0.428 (0.26, 0.571) for training with GAN-transformed images and 0.517 (0.366, 0.644) for predicting on transformed images. The registration-based approach achieves the highest Spearman correlation, which may indicate that local label information is preferable compared to weak labels. The performance of this approach may improve with better registration. All

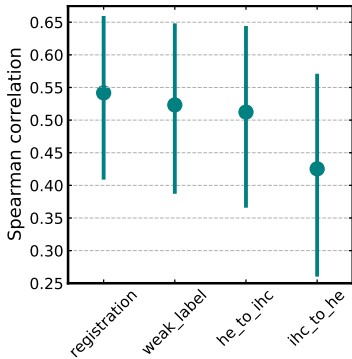

Figure 1: Bootstrapped 95% CI of Spearman correlations from 5-fold CV for training with registered or weak labels or Cycle-GAN stain transformations.

correlations are quite similar, except for training with generated images, which is in accordance with previous studies (Gupta et al., 2018). However, given this performance and the additional uncertainties of generated images, it can be questioned whether current CNN-based stain transformation techniques can be considered an appropriate solution for clinical settings if they do not clearly outperform existing alternatives. Our findings require validation in a larger cohort to meaningfully distinguish between the suggested methods with a held-out test set instead of CV, ideally with an analysis of clinical outcomes. If more data becomes available, it would also be possible to evaluate CNN models that aim to directly predict WSI-level scores, such as multiple instance learning techniques. In conclusion, our findings indicate that directly predicting Ki67 scores from H&E stained WSIs may offer a scalable solution for the prognostic stratification of ER-positive breast cancer patients.

## Acknowledgments

This project was supported by the Swedish Research Council, Swedish Cancer Society, ERA PerMed (ERAPERMED 2019-224-ABCAP), MedTechLabs, Swedish e-science Research Centre (SeRC).

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
