# OpenReview forum: "Prediction of Ki67 scores from H&E stained breast cancer sections using convolutional neural networks"
_MIDL.io/2021/Conference/Short — MIDL 2021 Poster_

### Official Review · Reviewer_BJCA · 2021-04-23

**Confidence:** 4
**Final Rating:** 3

**Summary:**

This paper aims to address the challenges of IHC-based Ki67 scoring in the clinical evaluation of breast cancer by predicting Ki67 scores from routine H&E stained whole-slide images.  Four different methods were evaluated in a cohort of 126 patients. The performance is reported as the Spearman correlation between predicted and IHC determined ROI-averaged K67 scores.

**Strengths:**

1. The paper address an important challenge in IHC-based K67 scoring, which could potentially have a significant impact on the clinical practice regarding breast cancer;

2. The paper investigated the performance of four different methodologies, which could provide an initial assessment on the future direction to pursue in this task.

**Weaknesses:**

1. Why is ROI *averaged* K67 scores used in calculating the Spearman's correlation? Could the authors calculate the correlation between per-tile prediction and ground truth instead? It would be interesting to see this performance on the first model, i.e., consider the slide-wise average score as a tile-wise weak label. It might still produce a good tile-wise prediction.
2. Regression task is relatively uncommon compare to classification tasks in machine learning literature. It might be helpful to provide a control, e.g., shuffle the H&E and ground truth independently and calculate the average Spearman's correlation.
3. What is the number of samples used for calculating the bootstrapped CI? For the purpose of comparing the methods, is it possible to train more models instead of using bootstrap? For example, do leave-one-out validation instead of 5-fold cross-validation. It might be helpful to provide a discussion on why the experiments are designed this way.
4. Have the authors consider recent publications on performing the same tasks? Such as this one
Liu, Yiqing, et al. "Predict Ki-67 positive cells in H&E-stained images using deep learning independently from IHC-stained images." Frontiers in Molecular Biosciences 7 (2020).
It may be helpful to discuss the relationship of the current work with previous works.

**Deanonymize Review:**

no

**Justification Of The Rating:**

The paper address an important challenge in predicting Ki67 scores in breast cancer. Despite the potential shortcomings mentioned in the Weakness section, I believe this paper is an interesting step in the particular research direction and could benefit the research community.

**Paper Type:**

both

**Special Issue:**

no

---

### Official Review · Reviewer_Voqy · 2021-05-06

**Confidence:** 5
**Final Rating:** 3

**Summary:**

The authors explore different strategies to calculate a Ki67 index using H&E images. The four approaches explored are: 1) using the Ki67 index as a weak label, 2) using labels of registered Ki67-stained tiles, 3) converting IHC to H&E and training an algorithm on those tiles and, conversely, 4) converting H&E to IHC and performing counting in IHC space. The methods show reasonable Spearman correlations coefficient to the reference established on IHC.

**Strengths:**

The authors consider very diverse methods to tackle the problem, using weakly supervised learning and cycle-GAN-based stain transformation. The dataset size is reasonable to tackle this problem, although I expect that especially for the weakly supervised case better performance would be achieved using more data. The paper does a decent job explaining the four different experiments and the use-case for this method.

**Weaknesses:**

The main question I would have is: why not simply do mitosis detection in the H&E? This is probably highly correlated to Ki67 index (both measure proliferation) and there is expansive literature on that topic. In terms of the paper, almost all details on networks used to address this problem are missing. I think the part on registration and introduction could have been shortened to add these details.

**Deanonymize Review:**

no

**Justification Of The Rating:**

The methodological approaches are interesting and offer some decent performance. The application itself is relevant. The missing details on how the algorithms were trained are less important in a short paper, but hopefully will be addressed at the conference.

**Paper Type:**

validation/application paper

**Special Issue:**

no

---

### Meta-Review · Program_Chairs · 2021-05-11

**Recommendation:** Accept (Poster)
**Confidence:** 5

**Metareview:**

This paper is a clear acceptance. Authors are suggested to address reviewer suggestions in final version.

---

### Decision · Program_Chairs · 2021-05-11

Accept (Poster)